# The Role of [^18^F]FDG PET and Clinicopathologic Factors in Detecting and Predicting Bone Marrow Involvement in Non-Hodgkin Lymphoma

**DOI:** 10.3390/cancers17020231

**Published:** 2025-01-13

**Authors:** Akram Al-Ibraheem, Ahmad Saad Abdlkadir, Nabil Hasasna, Hasan Alalawi, Ali Mohamedkhair, Salem Al-Yazjeen, Shahed Obeidat, Obayda Rabei, Dua’a Al-Jarrah, Omar Shahin, Kamal Al-Rabi, Mohammad Makoseh, Nidal Almasri

**Affiliations:** 1Department of Nuclear Medicine and PET/CT, King Hussein Cancer Center (KHCC), Al-Jubeiha, Amman 11941, Jordan; 2School of Medicine, University of Jordan, Al-Jubeiha, Amman 11942, Jordan; 3Department of Cell Therapy & Applied Genomics, King Hussein Cancer Center (KHCC), Al-Jubeiha, Amman 11941, Jordan; 4Department of Pathology, King Hussein Cancer Center (KHCC), Al-Jubeiha, Amman 11941, Jordan; 5Department of Medicine, King Hussein Cancer Center (KHCC), Al-Jubeiha, Amman 11941, Jordan

**Keywords:** [^18^F]FDG PET, lymphoma, bone marrow biopsy, positron emission tomography, NHL, bone marrow involvement

## Abstract

Numerous investigations have explored the diagnostic capability of [^18^F]fluorodeoxyglucose ([^18^F]FDG) positron emission tomography/computed tomography (PET/CT) in detecting bone marrow involvement in non-Hodgkin’s lymphoma. Most of these studies have utilized bone marrow biopsy as the sole reference standard to measure the diagnostic accuracy of [^18^F]FDG PET/CT. Despite its optimal specificity, bone marrow biopsy’s sensitivity is hampered by the limited scope and extent of bone marrow evaluation, as well as its traumatic sequelae. In our study, we aimed to evaluate [^18^F]FDG PET/CT diagnostic accuracy using two reference standards: bone marrow biopsy and clinical follow-up. A retrospective examination of 262 patients with various non-Hodgkin lymphoma subtypes yielded promising results, indicating that PET/CT has the potential to provide comprehensive skeletal evaluation and non-invasive assessment of bone marrow. One of the main strengths of this hybrid imaging modality is its ability to identify extra-iliac involvement beyond the scope of bone marrow biopsy.

## 1. Introduction

Bone marrow biopsy (BMB) is a commonly utilized procedure in the diagnostic workup of lymphoma patients [1]. The presence of lymphoma within the bone marrow can significantly impact disease staging, prognosis, and treatment decisions [2,3,4,5]. Despite its diagnostic value, BMB carries risks of pain, bleeding, and infection, and it is susceptible to sampling error if not conducted on an affected area of the marrow [6]. BMB typically targets the iliac crest due to its accessibility, safety, and rich hematopoietic tissue [7]. While alternative sites like the sternum exist, they pose higher risks and complexities and are seldom pursued [8]. In general, BMB has many limitations, including potential sampling errors and missed isolated distant bone marrow involvement (BMI). To mitigate these issues, complementary imaging techniques and multiple sampling sites may be necessary for comprehensive assessment [8].

Positron emission tomography (PET) using [^18^F]fluorodeoxyglucose ([^18^F]FDG) is a highly recommended and widely utilized imaging technique for staging non-Hodgkin lymphoma (NHL), providing reliable and precise results [9]. Recent interest has emerged in utilizing [^18^F]FDG PET findings as a supplemental or alternative method to BMB for evaluating BMI in NHL. A recent systematic review of previous studies exploring [^18^F]FDG’s utility for detecting BMI concluded that [^18^F]FDG PET/computed tomography (CT) can detect bone marrow lesions that may have been missed by BMB, making it a valuable complementary tool, even for indolent NHL subtypes [10]. Similar advancements have been made in the staging of Hodgkin lymphoma, where [^18^F]FDG PET/CT has largely replaced the need for formal BMB [11]. However, existing evidence primarily stems from studies conducted in developed regions, with limited representation of Arab or Middle Eastern populations.

The objective of this study is to report the results from a single cancer care center based in Jordan where both [^18^F]FDG PET and BMB were regularly conducted concurrently for staging patients with NHL over a 14-year timeframe from November 2005 to December 2019. This study examines the diagnostic accuracy and predictive capabilities of [^18^F]FDG PET, as well as clinical and biochemical factors.

## 2. Materials and Methods

This retrospective study involved the collection of data from new patients aged 14 years and older who were diagnosed with histopathology-proven NHL and staged at the King Hussein Cancer Center (KHCC) in Amman, Jordan, between November 2005 and December 2019. Institutional review board approval for this study was obtained from the Office of Human Research Protection Program at KHCC back in 29-July-2024 (registration number: 20 KHCC 156).

### 2.1. Data Collection

Specialized hematopathologists and nuclear medicine physicians independently extracted and analyzed electronic reports for BMB and [^18^F]FDG PET. Patient data, including demographics, clinical history, laboratory results, pathology findings, disease stage, and treatment outcomes, were collected from medical records and consultations with the lymphoma multidisciplinary team. The Ann Arbor system was used for NHL staging [12]. Baseline clinicopathologic data related to initial NHL staging were gathered from oncology clinic notes at the time of the initial NHL diagnosis. Data collection was performed using Microsoft Excel version 2021 (Redmond, WA, USA).

### 2.2. [^18^F]FDG PET/CT Imaging

PET/CT imaging was performed approximately 60 min after intravenous administration of 3–5 MBq/kg of [^18^F]FDG using a simultaneous Biograph mCT 64 PET/CT scanner (Siemens, Erlangen, Germany). Patients were required to fast for a minimum of 6 h prior to the scan, and all measured serum glucose levels were below 200 mg/dL. PET images were captured in 3D mode from the skull base to the mid-thigh using FlowMotion technology, with a table speed of 1 mm/second, equivalent to 3 min per bed position. PET image reconstruction was performed using the Ordered Subsets Expectation Maximization algorithm. Low-dose CT without intravenous contrast administration was employed for attenuation correction and anatomical localization.

### 2.3. [^18^F]FDG PET Scan Interpretation

Syngo.via Software (version VB40, Siemens, Erlangen, Germany) was utilized for the analysis of all [^18^F]FDG PET/CT images. Two experienced nuclear medicine physicians, blinded to the BMB findings and clinical follow-up data, independently reinterpreted the [^18^F]FDG PET data. Discrepancies were resolved through discussion to reach a consensus. The qualitative assessment of [^18^F]FDG PET images adhered to the methodology established by Lim et al. [13]. Normal bone marrow uptake was defined as homogeneous activity below liver intensity. Focal/multifocal uptake was characterized by one or more localized areas with uptake surpassing the liver reference, unexplained by CT or clinical correlation. Homogeneous uptake was defined as evenly distributed activity equal to or greater than the liver intensity, while heterogeneous uptake was defined as unevenly distributed activity throughout the bone marrow (Figure 1). The assessment of [^18^F]FDG PET positivity for bone marrow involvement was conducted through visual interpretation by nuclear medicine specialists, following the methodological framework established by Kaddu-Mulindwa et al. [14].

In instances of increased liver uptake due to lymphoma involvement, mediastinal blood pool activity was employed as the normal reference [13].

### 2.4. Definition of Bone Marrow Involvement

BMB served as the first reference standard. Bone marrow infiltration was assessed using a unilateral or bilateral iliac crest biopsy, with specimens analyzed by a reference hematopathologist. Lymphoma presence was determined through standard immunohistochemistry, utilizing antibodies to identify CD3-, CD79a-, and CD20-positive cells. A second reference standard, based on clinical follow-up data, was implemented to evaluate the incidence and nature of hypermetabolic bone marrow lesions. Bone marrow involvement (Table 1) was confirmed by either positive bone marrow histology or the resolution of bone marrow abnormalities on follow-up [^18^F]FDG PET/CT concurrent with a successful treatment response [15].

Both reference standards were employed to determine the diagnostic test accuracy of [^18^F]FDG PET/CT and its corresponding concordance rate.

### 2.5. Statistical Analysis

The normality of variables was evaluated through the Shapiro–Wilk W test. Normally distributed data are presented as the mean ± standard deviation (SD), while non-normally distributed data are presented as the median and interquartile range (IQR). Categorical data are displayed as frequencies and proportions. Sensitivity, specificity, and accuracy were calculated for both NHL as a whole and its subtypes, provided that a 2-by-2 contingency table could be constructed. The definition and methodology of sensitivity, specificity, and accuracy have already been highlighted and explained in previous publications by Šimundić et al. [16]. The McNemar test was used to examine significant statistical differences between two diagnostic test results [17]. The concordance rate between [^18^F]FDG PET and the standard reference was determined. Logistic regression analysis was employed to identify predictive factors for BMI. We used clinical follow-up data on bone marrow involvement as the dependent variable for both univariate and multivariate logistic regression analyses. Statistical significance was defined as a *p* value < 0.05. Stata software version 17 (Stata Corporation, College Station, TX, USA) was utilized for the statistical analysis.

## 3. Results

This retrospective study included 262 patients with NHL who underwent both [^18^F]FDG PET and BMB prior to initiating NHL therapy at our cancer center between November 2005 and December 2019. More than half of these patients were diagnosed with diffuse large B-cell lymphoma (DLBCL). Follicular lymphoma (FL) was the second most prevalent subtype, accounting for 18% of the cohort. The majority of patients were male (61.8%) and had advanced-stage NHL (stage III or IV) at diagnosis (70.6%). Additional baseline clinicopathologic metrics are detailed in Table 2.

### 3.1. Identification of Bone Marrow Involvement

BMI was detected in 71 NHL patients via BMB and in 81 patients using [^18^F]FDG PET. Clinical follow-up identified 104 patients with BMI. The concordance rate between [^18^F]FDG PET and BMB was 75.6%, and between BMB and clinical follow-up, it was 87.4%. Notably, the concordance rate between [^18^F]FDG PET and clinical follow-up was higher at 88.1%. The primary reason for discordance between [^18^F]FDG PET and BMB was the identification of extra-iliac focal hypermetabolic bone marrow lesions by [^18^F]FDG PET, which were negative on BMB in 30 patients (Figure 2).

Focal BMI patterns predominated, observed in 61 patients (23.3%). Additional etiologies for discordance along with important remarks are detailed in Table 3.

### 3.2. Diagnostic Test Evaluation

Using BMB as the reference standard, [^18^F]FDG PET demonstrated a sensitivity, specificity, and accuracy of 62.9%, 80%, and 75.6%, respectively. When clinical follow-up served as the reference, diagnostic accuracy improved significantly, with a sensitivity, specificity, and accuracy of 74.1%, 97.5%, and 88.2%, respectively (McNemar test; *p*-value < 0.0001). Notably, the overall detection rate (i.e., sensitivity) of BMB decreased from 100% to 68.3% when clinical follow-up was used as the standard reference. Table 4 presents the sensitivity, specificity, accuracy, and detection rates of diagnostic tests according to the two adopted reference standards.

### 3.3. Baseline [^18^F]FDG PET Implications for Clinical Management

Overall, 33 patients initially presenting with focal/multifocal extra-iliac bone marrow hypermetabolic lesions at the baseline [^18^F]FDG PET underwent treatment escalation based on [^18^F]FDG PET/CT imaging results, despite having negative BMB findings. This represents approximately 12.6% of the entire sample cohort.

### 3.4. Clinical, Biochemical, and Molecular Imaging Predictors of Bone Marrow Involvement

Univariate regression analysis demonstrated that focal/multifocal positive [^18^F]FDG PET imaging patterns exhibited statistically significant predictive value as molecular imaging biomarkers for BMI. Advanced-stage NHL (stage III or higher) served as a prognostic clinical indicator for BMI. Leukopenia and thrombocytopenia were identified as significant hematological parameters predictive of BMI. Furthermore, elevated levels of alkaline phosphatase and lactate dehydrogenase were significantly predictive of BMI. Subsequent multivariate analysis confirmed that focal/multifocal positive [^18^F]FDG PET imaging patterns, thrombocytopenia, and advanced NHL stage remained statistically significant independent predictors of BMI (Table 5).

## 4. Discussion

Our study highlights the diagnostic and predictive potential of [^18^F]FDG PET for evaluating BMI in NHL patients. Compared to BMB with clinical follow-up as the reference standard, [^18^F]FDG PET demonstrated greater sensitivity and concordance rates in identifying BMI. Significantly, the identification of hypermetabolic lesions outside the iliac bones on [^18^F]FDG PET scans, even in cases where BMB findings were negative, resulted in therapy escalation for approximately 13% of patients. A strong correlation was observed between the occurrence of focal [^18^F]FDG PET uptake patterns and BMI. Moreover, thrombocytopenia and advanced NHL stage were identified as supplementary indicators that may be used to predict BMI, highlighting the need to incorporate clinical, laboratory, and imaging data to ensure precise BMI evaluation.

The diagnostic utility of [^18^F]FDG PET/CT for detecting BMI in NHL patients remains a subject of ongoing debate. Recent studies have indicated that while [^18^F]FDG PET/CT provides valuable diagnostic information, its sensitivity may not be sufficient to entirely replace BMB [18,19,20,21,22]. A systematic review revealed that for NHL patients, [^18^F]FDG PET/CT demonstrated a sensitivity and specificity of 73% and 90%, respectively, in identifying BMI, compared to 56% and 100%, respectively, for identifying BMB [10]. These findings suggest that [^18^F]FDG PET/CT offers superior sensitivity but inferior specificity relative to BMB when using clinical follow-up data as a reference standard. Given this variability in diagnostic performance, [^18^F]FDG PET/CT is recommended as a complementary method rather than a replacement for BMB [10].

Our study revealed important findings about [^18^F]FDG PET scan diagnostic accuracy in NHL subtypes. The overall accuracy was 88.2% using clinical follow-up as a reference standard, but only 75.6% with BMB as a reference. Accuracy varied among NHL subtypes, being highest in DLBCL and FL, with lower values for mantle cell lymphoma (MCL), peripheral T-cell lymphoma (PTCL), and marginal zone lymphoma (MZL). These accuracy differences stem from varying FDG avidity among NHL subtypes. [^18^F]FDG expression depends on factors like tumor cell metabolism, cellular composition, proliferation rate, and histological grade [23]. More aggressive subtypes like DLBCL have higher metabolic activity and [^18^F]FDG uptake. The proportion of neoplastic to reactive inflammatory cells also affects uptake. For example, DLBCL has abundant tumor cells and shows mainly tumor-related FDG uptake [24]. Higher proliferation rates, often indicated by a higher Ki-67 index, generally mean increased FDG avidity. High-grade lymphomas typically show higher uptake than indolent, low-grade ones. For example, DLBCL and high-grade FL are highly [^18^F]FDG-avid, showing increased uptake due to their aggressive nature and high metabolic activity [9]. PTCL is generally [^18^F]FDG-avid, but uptake varies by specific subtype and tumor biology. MZL and MCL often show lower, more variable [^18^F]FDG uptake, possibly due to their less aggressive nature and lower metabolic activity [9].

Research specifically examining the utility of [^18^F]FDG PET in detecting isolated hypermetabolic lesions outside the iliac region is limited. However, several studies have reported beneficial results similar to our findings. Lee et al. conducted a retrospective analysis, identifying 11 NHL patients with baseline isolated distant extra-iliac hypermetabolic bone marrow lesions, negative iliac crest BMB, and confirmed BMI on clinical follow-up [25]. Schaefer et al. reported the beneficial diagnostic utility of [^18^F]FDG PET in two NHL patients who were negative for BMB but positive for focal hypermetabolic extra-iliac bone marrow lesions, which were confirmed to be lymphomatous upon direct bone biopsy [19]. In addition, Muslimani et al. documented five cases of extra-iliac localized hypermetabolic bone marrow lesions that were confirmed to be affected by NHL with direct bone biopsy, despite having negative results from iliac crest BMB [26]. These findings can significantly impact management plans, especially when a hypermetabolic extra-iliac lesion is the sole manifestation of extranodal NHL. This conclusion was supported in our research as well as in the earlier study performed by Kandeel et al. In their investigation, 12 out of the total sample cohort of 138 patients with NHL and Hodgkin lymphoma (8.6%) witnessed therapy escalation after the discovery of an extra-iliac hypermetabolic bone marrow lesion with discordant negative iliac crest BMB [27].

The evaluation of [^18^F]FDG PET/CT imaging patterns in BMI assessments presents a significant challenge, contributing to interrater variability. Not all hypermetabolic bone marrow localization should be automatically attributed to BMI [28]. Diffuse homogeneous uptake, in particular, requires careful clinical correlation to exclude reactive etiologies such as anemia and infection, which may overlap with NHL presentation [29]. Conversely, focal uptake patterns are generally more indicative of BMI, demonstrating greater reliability, as evidenced by our study and previous investigation by Berthet et al. [30].

In addition to [^18^F]FDG PET/CT, comprehensive clinical and biochemical evaluations are crucial for assessing BMI in NHL patients. Thrombocytopenia, for instance, has been associated with poor prognosis in patients with advanced NHL and poor BMI [31,32]. Aguado-Vázquez et al. conducted a study of 355 lymphoma patients, exploring clinical and biochemical predispositions for BMI. Their findings revealed that the presence of B symptoms, advanced lymphoma stage, neutropenia, and thrombocytopenia were independent predictive factors associated with BMI [33]. These results underscore the importance of adopting a holistic approach that integrates molecular imaging with clinical and laboratory parameters when evaluating the BMI of NHL patients.

Our study suffers from its retrospective nature, unicentric experience, and NHL heterogeneity. Nevertheless, it represents the largest series of Arab NHL patients of Middle Eastern decent, who have not been adequately studied previously. Our results add to the current evidence and support the use of clinical, biochemical, and molecular imaging utilities to diagnose and predict BMI.

## 5. Conclusions

[^18^F]FDG PET has both diagnostic and predictive utilities in evaluating BMI in NHL. Focal hypermetabolic extra-iliac bone marrow lesions indicate NHL involvement beyond the scope of BMB. Patients with such lesions, devoid of other extranodal sites, often require significant treatment intensification based on molecular imaging. Integrating molecular imaging with clinical and biochemical factors is crucial for accurate BMI prediction, underscoring the importance of multidisciplinary management. Our study is the first retrospective investigation on a large series of Arab NHL patients of Middle Eastern descent. A larger prospective study with a prospective design is still warranted to examine the wide-spectrum of [^18^F]FDG PET utilities in NHL patients.

## Figures and Tables

**Figure 1 cancers-17-00231-f001:**
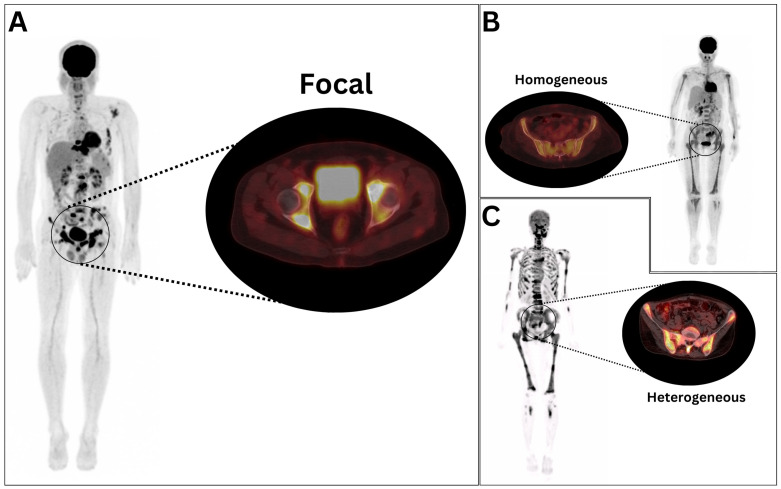
Maximum-intensity [^18^F]fluorodeoxyglucose (^18^F]FDG) positron emission tomography/computed tomography (PET/CT) images alongside axial pelvic views of representative non-Hodgkin lymphoma patients with (**A**) focal, (**B**) diffuse homogeneous, and (**C**) diffuse heterogeneous pattern of bone marrow uptake.

**Figure 2 cancers-17-00231-f002:**
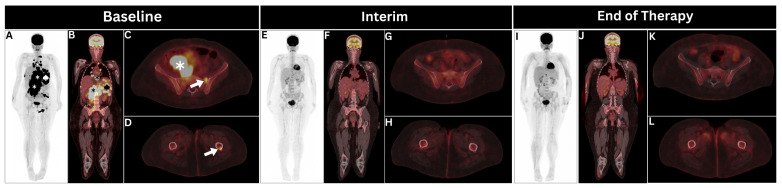
A 34-year-old female patient, suspected of having lymphoma, exemplifies the successful identification of extra-iliac multifocal lymphomatous bone marrow disease. (**A**–**D**) Maximum-intensity [^18^F]fluorodeoxyglucose (^18^F]FDG) positron emission tomography/computed tomography (PET/CT) projection images, along with coronal and axial views, revealed hypermetabolic bulky thoraco-abdomino-pelvic lymphadenopathy (asterisks) and multifocal splenic lesions (spades) alongside left sacral and left femoral shaft bone marrow lesions (arrows). Mediastinal lymph node biopsy confirmed the diagnosis of diffuse large B-cell lymphoma, while iliac crest bone marrow biopsy was negative. (**E**–**L**) The patient was offered six cycles of the Rituximab, Cyclophosphamide, Doxorubicin, Vincristine, and Prednisolone chemotherapy protocol (R-CHOP), which resulted in the complete metabolic resolution of lymphomatous disease on interim PET (after the third cycle) and at the end of therapy (after the sixth and final cycle). Utilizing follow-up PET/CTs for clinical follow-up revealed complete resolution of the hypermetabolic bone marrow lesions during the interim period and beyond, corresponding to the response of nodal hypermetabolic lesions, thus indicating initial positive bone marrow involvement based on clinical follow-up.

**Table 1 cancers-17-00231-t001:** Definition of bone marrow involvement.

Reference Standard	Criterion	Timeline	Additional Notes
Bone Marrow Biopsy	Positive histology showing lymphomatous infiltration	Initial diagnosis	Whether bilateral or unilateral
Clinical Follow-up	Hypermetabolic Extra-iliac bone marrow lesion(s)	Baseline [^18^F]FDG ^1^ PET/CT ^2^	Whether single or multiple
	Resolution of bone marrow abnormalities on follow-up [^18^F]FDG PET/CT	Interim [^18^F]FDG PET/CT or end of therapy [^18^F]FDG PET/CT	Concurrent with successful treatment response
	Persistent hypermetabolic bone marrow lesions	Interim [^18^F]FDG PET/CT or end of therapy [^18^F]FDG PET/CT	Alongside other sites of active disease
	Complete metabolic response in bone marrow	Interim [^18^F]FDG PET/CT or end of therapy [^18^F]FDG PET/CT	Alongside all other sites of active disease

^1^ [^18^F]FDG, [^18^F]fluorodeoxyglucose; ^2^ PET, positron emission tomography.

**Table 2 cancers-17-00231-t002:** Descriptive clinicopathological and demographic data for included patients at baseline timepoint.

Demographics	
Age At Diagnosis (in Years)	
Mean ± SD ^1^	49.6 ± 16.8 years
Gender (Number, Percentage)	
Male	162, (61.8%)
Female	100, (38.2%)
Clinical Characteristics (Number, Percentage)	
ECOG ^2^ Performance Status Scale	
ECOG 0–1	228, (87%)
ECOG ≥ 2	34, (13%)
B-Symptoms	86, (32.8%)
Laboratory Findings	
Biochemical Profile (Median & IQR ^3^, or Mean ± SD)	
Lactate Dehydrogenase (U/L)	252 (192–393)
Alkaline Phosphatase (U/L)	85 (70–107)
ESR ^4^ (mm/hr)	25 (12–51)
Hemoglobin (g/dL)	12.7 ± 2.2
Leukocyte (103/µL)	7.8 (6–9.9)
Platelet Count(103/µL)	257 (182–332)
Histopathological Characteristics	
NHL ^5^ Subtypes (Number, Percentage)	
DLBCL ^6^	137, (52.3%)
FL ^7^	47, (18%)
MCL ^8^	14, (5.3%)
PTCL ^9^	14, (5.3%)
MZL ^10^	13, (5%)
BCL ^11^	11, (4.2%)
ALCL ^12^	8, (3.1%)
BL ^13^	6, (2.3%)
SLL ^14^	5, (1.9%)
PbL ^15^	4, (1.5%)
HSTCL ^16^	2, (0.7%)
AITL ^17^	1, (0.4%)
Overall Initial Staging (Number, Percentage)	
Stage I	38, (14.5%)
Stage II	39, (14.9%)
Stage III	40, (15.3%)
Stage IV	145, (55.3%)

^1^ SD, standard deviation; ^2^ ECOG, Eastern Cooperative Oncology Group; ^3^ IQR, interquartile range; ^4^ ESR, erythrocyte sedimentation rate; ^5^ NHL, non-Hodgkin lymphoma; ^6^ DLBCL, diffuse large B-cell lymphoma; ^7^ FL, follicular lymphoma; ^8^ MCL, mantle cell lymphoma; ^9^ PTCL, peripheral T-cell lymphoma; ^10^; MZL, marginal zone lymphoma; ^11^ BCL, B-cell lymphoma; ^12^ ALCL, anaplastic large cell lymphoma; ^13^ BL, Burkitt’s lymphoma; ^14^ SLL, small lymphocytic lymphoma; ^15^ PbL, plasmablastic lymphoma; ^16^ HSTCL, hepatosplenic T-cell lymphoma; ^17^ AITL, angioimmunoblastic T-cell lymphoma.

**Table 3 cancers-17-00231-t003:** Table summarizing the frequency and [^18^F]fluorodeoxyglucose ([^18^F]FDG) positron emission tomography (PET) imaging patterns for bone marrow lesions.

Bone Marrow Involvement (Number, Percentage)				
Per bone marrow biopsy results	71, (27.1%)			
Per clinical follow-up	104, (39.7%)			
Per [^18^F]FDG ^1^ PET ^2^ findings	81, (30.9%)			
Concordance Rate (Percentage)				
Between [^18^F]FDG PET and bone marrow biopsy	75.6%			
Between [^18^F]FDG PET and clinical Follow-up	88.1%			
Between bone marrow biopsy and clinical Follow-up	87.4%			
Patterns of bone marrow expression as depicted by [^18^F]FDG PET (Number, Percentage)				
Unremarkable for [^18^F]FDG expression	152, (58%)			
Focal/multifocal [^18^F]FDG expression	61, (23.3%)			
Homogeneous [^18^F]FDG expression	35, (13.4%)			
Heterogeneous [^18^F]FDG expression	14, (5.3%)			
Clinical Follow-up as a reference				
Etiology	(*n*, %)	BM pattern on [^18^F]FDG PET	False interpreter	Observed in (NHL ^3^ subtype, *n*)
Identification of extra-iliac focal hypermetabolic bone marrow lesions by [^18^F]FDG PET deemed negative on bone marrow biopsy	30, (11.5%)	Focal	Bone marrow biopsy	DLBCL ^4^ (*n* = 18); FL ^5^ (*n* = 7); ALCL ^6^ (*n* = 1); BL ^7^ (*n* = 1); BCL ^8^ (*n* = 1); MCL ^9^ (*n* = 1); MZL ^10^ (*n* = 1)
Unremarkable [^18^F]FDG PET study for bone marrow involvement deemed positive on bone marrow biopsy	16, (6.1%)	Unremarkable bone marrow pattern	[^18^F]FDG PET	DLBCL (*n* = 7); MCL (*n* = 3); MZL (*n* = 3); PTCL ^11^ (*n* = 2); FL (*n* = 1)
Widespread heterogeneous bone marrow expression on [^18^F]FDG PET deemed positive on bone marrow biopsy	8, (3.1%)	Diffuse homogeneous	[^18^F]FDG PET	FL (*n* = 4); ALCL (*n* = 1); FL (*n* = 1); PTCL (*n* = 1); SLL ^12^ (*n* = 1)
Widespread heterogeneous bone marrow expression on [^18^F]FDG PET deemed negative on bone marrow biopsy	7, (2.6%)	Heterogeneous vertebral	Bone marrow biopsy	DLBCL (*n* = 5); FL (*n* = 1); PTCL (*n* = 1)
Widespread heterogeneous bone marrow expression on [^18^F]FDG PET deemed positive on bone marrow biopsy	3, (1.1%)	Widespread heterogeneous	[^18^F]FDG PET	SLL (*n* = 2); MCL (*n* = 1)

^1^ [^18^F]FDG, [^18^F]fluorodeoxyglucose; ^2^ PET, positron emission tomography; ^3^ NHL, non-Hodgkin lymphoma; ^4^ DLBCL, diffuse large B-cell lymphoma; ^5^ FL, follicular lymphoma; ^6^ ALCL, anaplastic large cell lymphoma; ^7^ BL, Burkitt’s lymphoma; ^8^ BCL, B-cell lymphoma; ^9^ MCL, mantle cell lymphoma; ^10^; MZL, marginal zone lymphoma; ^11^ PTCL, peripheral T-cell lymphoma; ^12^ SLL, small lymphocytic lymphoma.

**Table 4 cancers-17-00231-t004:** Sensitivity, specificity, accuracy, and detection rates of diagnostic tests according to two adopted reference standards.

Bone Marrow Biopsy as a Reference Standard: For [^18^F]FDG ^1^ PET ^2^				
NHL ^3^	Sensitivity (95% CI ^4^)	Specificity (95% CI)		Accuracy (95% CI)
Overall	62.9% (50–74.2%)	80% (73.7–74.2)%		75.6% (69.9–80.6%)
DLBCL ^5^	74.1% (53.7–88.9%)	74.4% (64.1–83.1%)		74.4% (65.5–82%)
FL ^6^	68.7% (41.3–88.9%)	74.2% (55.4–88.1%)		72.3% (57.4–84.4%)
MCL ^7^	42.9% (9.9–81.6%)	71.4% (29–96.3%)		57.1% (28.9–82.3%)
PTCL ^8^	33.3% (0.8–90.6%)	90.9% (58.7–99.8%)		78.6% (49.2–95.3%)
MZL ^9^	25% (0.6–80.6%)	77.8% (39.9–97.2%)		61.5% (31.6–86.1%)
Clinical follow-up as a reference standard: For [^18^F]FDG PET				
NHL	Sensitivity (95% CI)	Specificity (95% CI)		Accuracy (95% CI)
Overall	74.1% (64.5–82.1%)	97.5% (93.6–99.3%)		88.2% (93.6–91.8%)
DLBCL	85.4% (72.2–93.9%)	97.5% (91.3–99.7%)		93% (87.1–96.7%)
FL	78.3% (56.3–92.5%)	95.8% (78.9–99.9%)		87.3% (74.3–95.2%)
PTCL	25% (0.6–80.6%)	90% (55.5–99.7%)		71.4% (74.9–91.6%)
Bone marrow biopsy Detection Rate				
NHL	Bone marrow biopsy referenced		Clinical follow-up referenced (95% CI)	
Overall	100%		68.3% (58.4–77%)	
DLBCL	100%		56.2% (41.2–705%)	
FL	100%		69.6% (47.1–86.8%)	

^1^ [^18^F]FDG, [^18^F]fluorodeoxyglucose; ^2^ PET, positron emission tomography; ^3^ NHL, non-Hodgkin lymphoma; ^4^ CI, confidence interval; ^5^ DLBCL, diffuse large B-cell lymphoma; ^6^ FL, follicular lymphoma; ^7^ MCL, mantle cell lymphoma; ^8^ PTCL, peripheral T-cell lymphoma BCL, ^9^ MZL, marginal zone lymphoma.

**Table 5 cancers-17-00231-t005:** Clinicopathologic, biochemical, and molecular imaging predictors of bone marrow involvement.

Univariate Analysis					
Factor	Reference	Category	Odds ratio	95% CI ^1^	*p*-value
NHL ^2^ Stage	Stage ≤ 2	Stage ≥ 3	10.7	4.7–24.6	0.00001
B Symptoms	Absence	Presence	1	0.6–1.7	0.96
ECOG ^3^ Score	ECOG < 2	ECOG ≥ 2	1.7	0.8–3.4	0.17
ALP ^4^ levels	<147 U/L	147 U/L	2.9	1.4–6.2	0.004
LDH ^5^ levels	<280 U/L	280 U/L	1.9	1.1–3.2	0.015
ESR ^6^ levels	Normal	Elevated	0.6	0.3–1.1	0.074
Hb ^7^ Levels	Normal	Anemia	0.6	0.3–1	0.07
WBC ^8^ Count	Normal	Leukopenia	2.7	1.4–5.3	0.003
PLT ^9^ Count	PLT ≥ 150 (103/µL)	PLT < 150	8.5	3.7–19.5	0.00001
BM ^10^ Pattern on [^18^F]FDG ^11^	Diffuse or unremarkable	Focal or multifocal	46.7	16.1–135.4	0.000001
**Multivariate Analysis**					
Factor	Reference	Category	Odds ratio	95% CI	*p*-value
NHL Stage	Stage ≤ 2	Stage ≥ 3	6.1	1.8–19.6	0.003
ALP levels	<147 U/L	147 U/L	2.2	0.7–7	0.17
LDH levels	<280 U/L	280 U/L	1.1	0.4–2.6	0.89
WBC Count	Normal	Leukopenia	2.3	0.8–6.5	0.13
PLT Count	PLT ≥ 150 (103/µL)	PLT < 150	9.4	3–29.8	0.00001
BM Pattern	Diffuse or unremarkable	Focal or multifocal	77.3	20.2–296.1	0.000001

^1^ CI, confidence interval; ^2^ NHL, non-Hodgkin lymphoma; ^3^ ECOG, Eastern Cooperative Oncology Group performance score; ^4^ ALP, alkaline phosphatase; ^5^ LDH, lactate dehydrogenase; ^6^ ESR, erythrocyte sedimentation rate; ^7^ HB, hemoglobin; ^8^ WBC, white blood cell; ^9^ PLT, platelet; ^10^; BM, bone marrow; ^11^ [^18^F]FDG, [^18^F]fluorodeoxyglucose.

## Data Availability

The data presented in this study are available on request from the corresponding author. The data are not publicly available due to privacy.

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
