# Peer review of "The Role of [^18^F]FDG PET and Clinicopathologic Factors in Detecting and Predicting Bone Marrow Involvement in Non-Hodgkin Lymphoma"

_cancers, 2025, doi:10.3390/cancers17020231_

Round 1

Reviewer 1 Report

Comments and Suggestions for Authors

The value of FDG PET positivity or negativity as a surrogate of pos/neg BM invasion in order to not perform BMB is well established.
The present retrospective study unfortunately did not add value to the topic.

Author Response

Dear Reviewer 1,

Thank you for your insightful feedback. It is very motivating to know that our work is appreciated and will hopefully have a positive impact on such an important topic.

Reviewer 2 Report

Comments and Suggestions for Authors

In this manuscript, by studying a large cohort of lymphoma patients the authors clearly describe the diagnostic and predictive role of FDG-PET in assessing bone marrow involvement (BMI), compared to bone marrow biopsy (BMB) and clinical follow-up.

This study sounds scientifically correct, confirms the already known superiority of FDG-PET over BMB in detecting BMI and identifies other clinical and laboratory parameters as significant predictors of BMI.

Reference section is adeguate.                 

Author Response

Dear Reviewer 2,

Thank you for your comments. It was very inspiring to recommend our work. We hope that our results would substantiate the value of [18F]FDG in this domain.

Reviewer 3 Report

Comments and Suggestions for Authors

The authors conducted a retrospective study to evaluate the diagnostic performance of FDG PET in identifying bone marrow involvement in non-Hodgkin lymphoma (NHL). Their findings indicate that FDG PET shows promise in this clinical context. However, the utility of FDG PET for detecting bone marrow involvement in NHL has been a subject of debate (Semin Nucl Med. 2023 May;53(3):320-351. doi: 10.1053/j.semnuclmed.2022.11.001.). Consequently, this study holds significant clinical relevance. My comments and suggestions are listed as follows,

1. The authors have demonstrated three major abnormal FDG PET marrow patterns in Figure 1. The patterns of their PET data are summarized in Table 3. However, the methods section did not clearly mention the definitions of a positive FDG PET marrow involvement. Please list the definitions of positive bone marrow involvement on FDG PET so the readers can follow the same methodologies.

2. The authors provided the staging of all included patients. Is the staging of bone marrow based on bone marrow biopsy, FDG PET, or both modalities? If a patient had a negative bone marrow biopsy but focal FDG uptake other than iliac bone, would this patient deemed as stage IV?

3. In results section 3.3. The authors mentioned that 33 patients had negative marrow biopsies but extra-iliac marrow FDG avid lesions. They were treated as having positive marrow involvements (treatment escalation). What were the survivals of these patients? Were the survival outcomes of these patients similar to those of patients with positive marrow biopsy?

4. The authors tested the association of many variables with bone marrow involvement using logistic regression. Was the reference standard bone marrow biopsy or clinical follow-up? 

5. The diagnostic performance of FDG PET is higher for those patients with more aggressive types of NHL. This study also demonstrated similar results. Please have a discussion of this issue. The differences in the diagnostic performance and the possible clinical utility of FDG PET in detecting bone marrow involvement among different histopathological types of NHL.

Author Response

Dear Review 3,
We truly appreciated your respectful feedback and would like to express our sincere gratitude for providing the time and effort to review this research work.

Below are point-by-point answers to your respectful review points

  1. The authors have demonstrated three major abnormal FDG PET marrow patterns in Figure 1. The patterns of their PET data are summarized in Table 3. However, the methods section did not clearly mention the definitions of a positive FDG PET marrow involvement. Please list the definitions of positive bone marrow involvement on FDG PET so the readers can follow the same methodologies.
    • We have outlined our adopted methodology for PET positivity in the amended version. Kindly track changes in lines 120-123 (Gray highlights).

  2. he authors provided the staging of all included patients. Is the staging of bone marrow based on bone marrow biopsy, FDG PET, or both modalities? If a patient had a negative bone marrow biopsy but focal FDG uptake other than iliac bone, would this patient deemed as stage IV?
    • We have clarified this important information in the revised version. Kindly track changes in lines 96-97 (Gray highlights).

  3. In results section 3.3. The authors mentioned that 33 patients had negative marrow biopsies but extra-iliac marrow FDG avid lesions. They were treated as having positive marrow involvements (treatment escalation). What were the survivals of these patients? Were the survival outcomes of these patients similar to those of patients with positive marrow biopsy?
    • Your suggestion is much appreciated, but analyzing survival impact would require new datasets and different analytical approaches. This deviates from our current focus on diagnosis. While intriguing, we prefer to reserve such analysis for a future project to maintain our current scope and objectives.

  4. The authors tested the association of many variables with bone marrow involvement using logistic regression. Was the reference standard bone marrow biopsy or clinical follow-up?
    • This was outlined in line 155-158 (Gray highlights).

  1. The diagnostic performance of FDG PET is higher for those patients with more aggressive types of NHL. This study also demonstrated similar results. Please have a discussion of this issue. The differences in the diagnostic performance and the possible clinical utility of FDG PET in detecting bone marrow involvement among different histopathological types of NHL.
    • This was highlighted within the Discussion section. Kindly track changes in lines 269-285 (Gray highlights).

Round 2

Reviewer 3 Report

Comments and Suggestions for Authors

Most of my suggestions and comments have been addressed.

Although adding survival analysis would further improve the overall merits of this study, I agree that survival analysis is not the main focus of this research.

I hope the authors will perform and report the survival analysis data in the future.